# Association of early viral lower respiratory infections and subsequent development of atopy, a systematic review and meta-analysis of cohort studies

**Sebastien Kenmoe[1], Cyprien Kengne-Nde[2], Abdou Fatawou Modiyinji[1,3], Jean Joel Bigna[4], Richard Njouom[1]***

**1** Department of Virology, Centre Pasteur of Cameroon, Yaoundé, Cameroon, **2** National AIDS Control Committee, Epidemiological Surveillance, Evaluation and Research Unit, Yaounde, Cameroon, **3** Department of Animals Biology and Physiology, Faculty of Sciences, University of Yaoundé I, Yaoundé, Cameroon, **4** Department of Epidemiology and Public Health, Centre Pasteur of Cameroon, Yaoundé, Cameroon

* njouom@pasteur-yaounde.org

**Data Availability Statement:** All relevant data are within the manuscript and its Supporting Information files.

## Abstract

### Introduction

Existing evidence on the relationship between childhood lower respiratory tract infections (LRTI) and the subsequent atopy development is controversial. We aimed to investigate an association between viral LRTI at <5 years and the development of atopy at > 2 years.

### Methods

We conducted a search at Embase, Pubmed, Web of Science, and Global Index Medicus. We collected data from the included articles. We estimated the odds ratio and the 95% confidence intervals with a random effect model. We determined factors associated with atopy development after childhood LRTI using univariate and multivariate meta-regression analyses. We recorded this systematic review at PROSPERO with the number CRD42018116955.

### Results

We included 24 studies. There was no relationship between viral LRTI at <5 years and skin prick test-diagnosed-atopy (OR = 1.2, [95% CI = 0.7–2.0]), unknown diagnosed-atopy (OR = 0.7, [95% CI = 0.4–1.3]), atopic dermatitis (OR = 1.2, [95% CI = 0.9–1.6]), hyperreactivity to pollen (OR = 0.8, [95% CI = 0.3–2.7]), food (OR = 0.8, [95% CI = 0.3–2.5]), or house dust mite (OR = 1.1, [95% CI = 0.6–2.2]). Although not confirmed in all studies with a symmetric distribution of the 23 confounding factors investigated, the overall analyses showed that there was a relationship between childhood viral LRTI at < 5 years and serum test diagnosed-atopy (OR = 2.0, [95% CI = 1.0–4.1]), allergic rhinoconjunctivitis (OR = 1.7, [95% CI = 1.1–2.9]), hyperreactivity diagnosed by serum tests with food (OR = 5.3, [1.7–16.7]) or

**Funding:** The author(s) received no specific funding for this work.

**Competing interests:** The authors have declared that no competing interests exist.

inhaled allergens (OR = 4.2, [95% CI = 2.1–8.5]), or furred animals (OR = 0.6, [95% CI = 0.5–0.9]).

## Conclusion

These results suggest that there is no association between viral LRTI at < 5 years and the majority of categories of atopy studied during this work. These results, however, are not confirmed for the remaining categories of atopy and more particularly those diagnosed by serum tests. There is a real need to develop more accurate atopy diagnostic tools.

## Introduction

Atopy is a genetic predisposition to the development of allergic diseases such as atopic dermatitis, atopic eczema, atopic asthma, atopic conjunctivitis or allergic rhinitis [1]. Atopy also includes increased hypersensitivity to inhaled or food allergens, with the development of IgE mediated by Th2 cells [2]. Atopic diseases is associated to a significant morbidity and a very important economic burden for society[3]. Atopic disease prevalence has experienced in recent decades an exponential increase in the world [4,5].

Common viruses associated with lower respiratory tract infections (LRTI) include Influenza, Rhinovirus, Respiratory Syncytial Virus (HRSV), Metapneumovirus, Parainfluenzavirus, Enterovirus, Adenovirus, Bocavirus, and Coronavirus [6,7]. Data have showed the association between HRSV LRTI and subsequent wheezing or asthma [8–11]. With the advent of molecular assays, the description of childhood infections caused by non-HRSV has further demonstrated the implication of these diseases in long-term sequelae [12,13]. A meta-analysis by Liu et al., have demonstrated the association between childhood RV infections and the subsequent development of asthma [13]. A systematic review showed that pneumonia mainly due to Adenovirus was linked to sequelae including obstructive pulmonary disease or chronic bronchitis [12]. The subgroup analyses in this latter however found that the 3 included studies with Mycoplasma pneumoniae pneumonia were not associated with long-term sequelae [12].

Studies on the prevalence of atopy among people presenting with viral LRTI in childhood have shown divergent results [14–16]. Some studies have reported an increased risk of allergic sensitization after viral LRTI in childhood [14,17]. Protection against allergic sensitization through the stimulation of Th-1 cytokine production has been suggested by other studies [16]. Maximum confusion has been demonstrated by other studies that have shown no influence of childhood viral LRTI in the development risk of subsequent atopy [15,18,19].

The resolution of the question of the association between viral LRTI in childhood and the subsequent development of atopy could serve as a basis for preventive measures and management of atopic diseases [20–22]. The purpose of this systematic review and meta-analysis of Long-term sEquelAe of lower Respiratory tract infections iN Early childhood (A LEARNED study) was to investigate the association between viral LRTI at <5 years and the atopy development at > 2 years.

## Methods

### Research design

This systematic review was registered in PROSPERO (Registration number: CRD42018116955). The study was conducted following the Centre for Reviews and

Dissemination guidelines [23] and reported according to the PRISMA (Preferred reporting items for systematic review and meta-analysis) guidelines (S1 Table) [24].

## Inclusion criteria

Study participants were children with a history of laboratory confirmed viral LRTI before 5 years. Viral LRTI was considered using the definition proposed by the authors of the included studies. Children with viral LRTI, as reference cases, were compared to control children who had no history of LRTI in childhood. Studies including only participants with medical conditions (premature birth, immunodeficiency or other comorbidities) were excluded. The exposure in this systematic review was a viral LRTI in children at <5 years. The outcome of this systematic review was the development of atopy at> 2 years including atopic diseases and sensitization to food and atmospheric allergens. Atopic status was determined by skin prick tests (SPT) and total or allergen-specific serum IgE antibody assessed by immunoassays. Included studies were prospective and retrospective cohorts with a minimum follow-up duration of one year. Atopic diseases were diagnosed clinically. We considered atopy category whose data on outcomes were available in three or more studies.

## Online search strategy and study selection

The Pubmed, Excerpta Medica Database, Web of Science, and Global Index Medicus databases were queried for articles published from inception through July 15, 2019. All languages and geographic areas were considered for this systematic review. The combination of terms used for the bibliographic search is listed in Table 1. We manually screened the included studies and relevant review reference lists to locate additional articles. Two independent investigators (KS and AFM) reviewed the titles and abstracts of the articles found by the electronic and manual search [25]. We summarized the selection process of potentially relevant articles on a PRISMA flowchart. The disagreements between the two investigators were resolved by discussion and consensus.

## Data extraction

Two researchers (KS and AFM) independently extracted data from full text of included articles. The following data was collected (title, first author, year of publication, time of data collection, country, participants interview period, LRTI type, LRTI rank, LRTI period, age at LRTI, virus associated with the LRTI, control age, control gender, total number of cases and controls,

**Table 1. Search strategy in Pubmed.**

| Field | Key words |
|---|---|
| #1 (Atopy) | Atop* OR Allerg* OR Asthma* OR hypersensitivity OR "immunoglobulin E" OR "Ig E" |
| #2 (LRTI) | LRTI OR ALRTI OR "Lower Respiratory Tract Infections" OR ALRI OR "Acute Lower Respiratory Infections" OR "Acute Lower Respiratory Tract Infections" OR SARI "Severe Acute Respiratory Infections" OR "Severe Acute Respiratory Illness" OR Bronchiolitis OR Pneumonia |
| #3 (Virus) | Virus* OR "viral infect*" OR HRSV OR RSV OR "Human Respiratory Syncytial Virus" OR "Respiratory Syncytial Virus" OR HMPV OR MPV OR "Human Metapneumovirus" OR Metapneumovirus OR HAdV OR AdV OR Adenovirus OR "Human Adenovirus" OR HBoV OR BoV OR Bocavirus OR HCoV OR CoV OR Coronavirus OR 229E OR OC43 OR NL63 OR HKU1 OR HPIV OR Parainfluenzavirus OR PIV-1 OR PIV-2 OR PIV-3 OR PIV-4 OR HPIV-1 OR HPIV-2 OR HPIV-3 OR HPIV-4 OR Enterovirus OR Coxsackievirus OR Echovirus OR Parechovirus OR Rhinovirus OR Rhinoviruses OR Influenza |
| #4 | #1 AND #2 AND #3 |

numbers with atopy at follow up, and data on confounders). The discrepancies were resolved by discussion and a consultation of a third arbitrator (RN) if necessary.

### Quality assessment of included studies

In accordance with Newcastle-Ottawa Scale (NOS) criteria including patient selection, comparability of groups, and outcome evaluation, two independent researchers (KS and AFM) assessed the quality of all included studies [26] (S2 Table). To reach consensus, all the differences were discussed between the two researchers.

### Data synthesis and analysis

We estimated Odds ratio (OR) as a measure of the association between childhood viral LRTI and subsequent atopy development. We evaluated the publication bias by visual inspection of funnel diagram and Egger test. We estimated heterogeneity between studies by the Q test and the $I^2$ statistic [27,28]. We considered heterogeneity as significant between studies for p-value $<0.1$ or $I^2 > 50\%$. We conducted sensitivity analyses with studies with a low risk of bias, studies including only inpatients or the first episode of viral LRTI. We performed subgroup analyses on the basis of the type of LRTI, WHO region, age at LRTI, age at follow up, and type of virus detected in LRTI. We applied a multivariate metaregression with a stepwise manual selection procedure to identify factors associated with the variation of overall risk of atopy. We successively removed from the model variables by considering the signification of the p-value and information criteria for the model like log-likelihood, deviance, Akaike information criterion (AIC), Bayesian information criterion (BIC), and corrected Akaike Information Criterion (AICc). We reported the explained heterogeneity (R2) by variables included in models. A variable with P value $< 0.05$ was considered statistically significant in the final model. We assessed the influence of confounding factors by conducting a sensitivity study that included only equitably distributed studies for each risk factor between reference cases and controls. For each potential confounding factor, we assessed the distribution between reference cases and controls by recalculating the p values using the exact tests of Fisher and Chi-2. The two p values $> 0.05$ of Fisher's exact test and Chi-2 indicated a symmetric distribution of the confounding factor between reference cases and controls.

## Results

### Literature search and characteristics of included studies

We synthesized the study selection process in Fig 1. The electronic (4634 articles) and manual (23 articles) searches identified 4657 articles. The first selection by titles and abstracts resulted in the exclusion of 4249 irrelevant articles. We read and fully reviewed the complete texts of the remaining 330 articles. We excluded a total of 309 articles for multiple reasons including mismatch of the study population (no control group, inclusion of non-viral LRTI and non-LRTI infections, and inclusion of only patients with underlying medical conditions), the type of study not appropriate (case report, comment on study, editorial, and review), lack of data on outcomes, conference abstract or complete texts not found, and irrelevance of the article-Supplementary. We finally included 22 articles (24 studies) in the qualitative and quantitative synthesis of this systematic review [29–50]. We showed the individual characteristics of the publications included in S3 Table. Most studies were conducted in Europe, detected HRSV, had a low risk of bias, included hospitalized children under 1 year of age with their first episode of bronchiolitis, had followed children between 5–10 years old, and were prospective. Children with a history of viral LRTI in childhood were recruited between 1960 and 2014 and articles

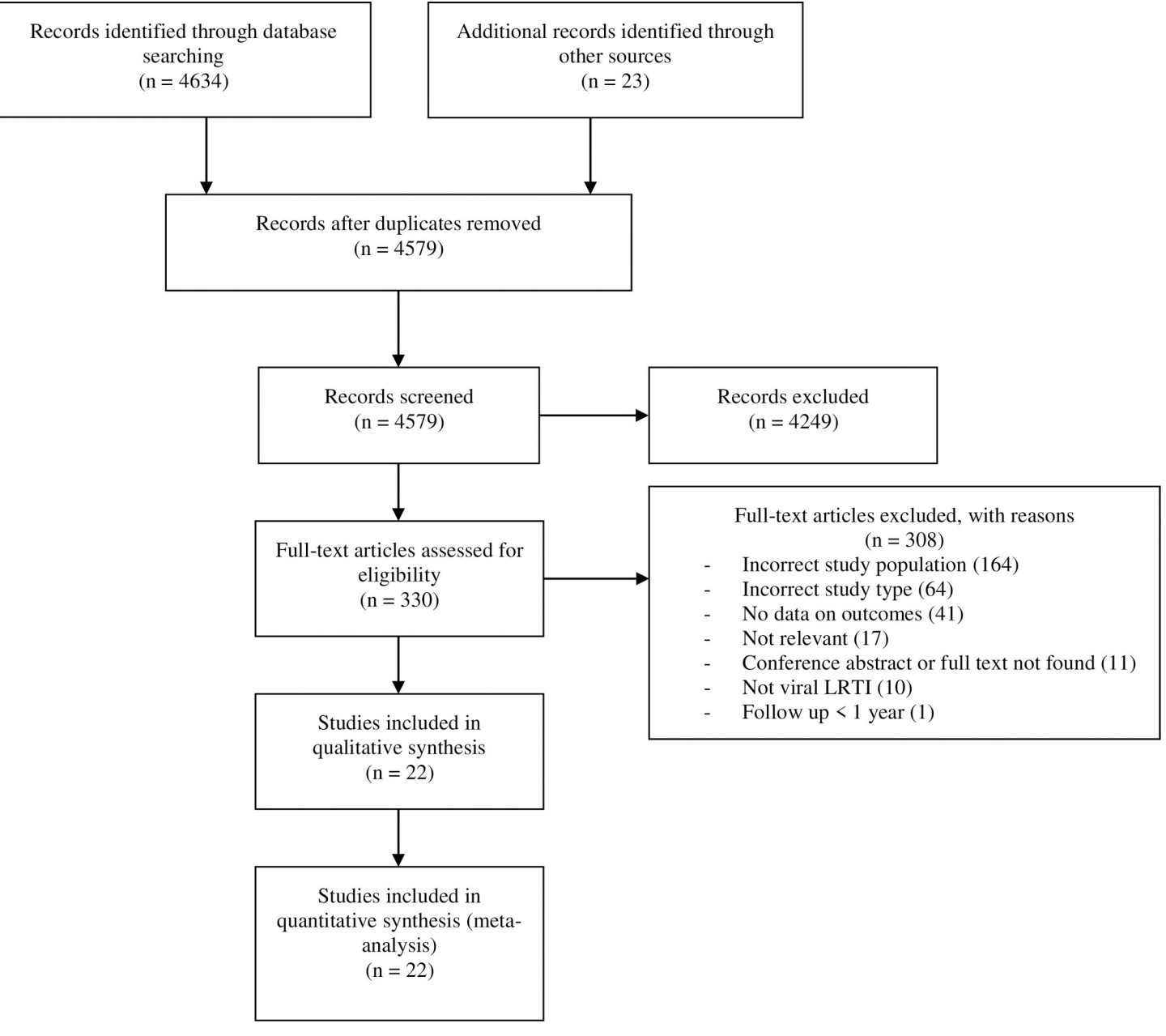

**Fig 1. Study selection.**

were published between 1981 and 2017.The individual NOS score from the included studies are presented in S4 Table.

## Comparison of reference cases with controls

The frequency of post-LRTI atopy was similar between reference cases and controls in most categories (atopy diagnosed with SPT, OR = 1.2, 95% CI = 0.7–2.0; atopy diagnosis unknown/not reported, OR = 0.7, 95% CI = 0.4–1.3; atopic dermatitis, OR = 1.2, 95% CI = 0.9–1.6; pollens, OR = 0.8, 95% CI = 0.3–2.7; food allergy, OR = 0.8, 95% CI = 0.3–2.5; and house dust mite, OR = 1.1, 95% CI = 0.6–2.2) (Fig 2, S1 Fig). With regard to atopy assess by serum test, we

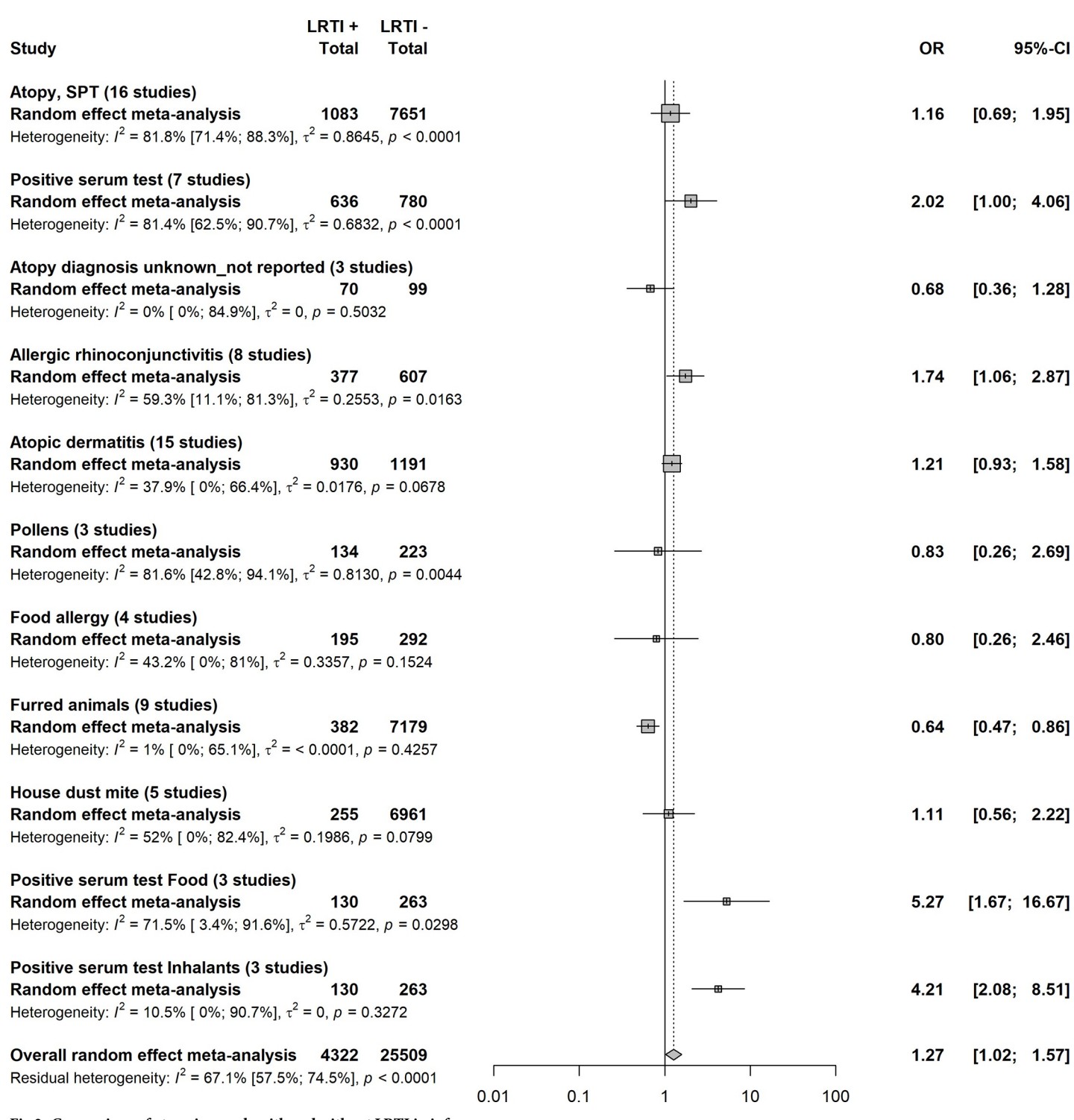

**Fig 2. Comparison of atopy in people with and without LRTI in infancy.**

observed significant differences in favor of reference cases including positive serum test (OR = 2.0, 95% CI = 1.0–4.1), food positive serum test (OR = 5.3, 95% CI = 1.7–16.7), and inhalant positive serum test (OR = 4.2, 95 CI % = 2.1–8.5). In 8 studies, allergic

rhinoconjunctivitis was significantly more frequently reported in reference cases than in controls (OR = 1.7, 95% CI = 1.0–2.9). Positivity for furred animals was significantly more frequent in controls compared to reference cases (OR = 0.6, 95% CI = 0.5–0.9). The overall effect remained unchanged for the majority of our results when assessed by sensitivity analyses of the impact of LRTI rank, hospitalization, and study quality (Table 2). The effect observed in the main analysis of the positive serum test was lost for studies reporting the first episodes of LRTI (OR = 2.0, 95% CI = 0.6–6.7) and hospitalized children (OR = 2.5, 95% CI = 0.9–6.5). In contrast to the association observed between LRTI history and development of allergic rhinoconjunctivitis (OR = 1.7, 95% CI = 1.1–2.9), no effect was observed for studies reporting the first episode of LRTI (OR = 1.4, 95% CI = 0.5–3.6). The significant preponderance of furred animal positivity in controls compared to reference cases was lost in studies including only the first LRTI episodes (OR = 0.5, 95% CI = 0.3–1.0).

## Subgroup analyses and metaregression

In the subgroup analyses (S5 Table), a statistically significant association between childhood viral LRTI and subsequent atopy was observed only in the bronchiolitis subgroup for the categories of atopy diagnosed by serum test (OR = 3.1, 95% CI = 2.0–4.8; p < 0.001), and allergic rhinoconjunctivitis (OR = 2.3, 95% CI = 1.3–3.9; p = 0.021). Atopy diagnosed by SPT was in favor of controls in retrospective studies (OR = 0.6, 95% CI = 0.5–0.8, p = 0.014). The difference in the development of atopy by age group at the time of LRTI development was statistically significant for atopy diagnosed by serum tests (p = 0.002) and positive serum test for food (p = 0.019). Children with LRTI at <9 months were at increased risk for atopy diagnosed by serum tests (OR = 20.5, 95% CI = 4.4–95.9) and positive serum test for food (OR = 41.5, 95% CI = 5.2–330.0). The development of atopy according to the age of the patients varied significantly for atopy diagnosed by SPT (p = 0.004), atopy diagnosed by serum test (p = 0.004), positivity for house dust mite allergen (p = 0.046), and for positive serum test for food (p = 0.030). Atopy positivity by SPT varied transiently with an association between 2 and 5 years (OR = 3.3, 95% CI = 1.9–5.6) and then between 15 and 20 years (OR = 1.6, 95% CI = 1.1–2.4). Positivity to atopy by serum tests was not significantly associated only for patients 5 to 10 years (OR = 1.0, 95% CI = 0.5–2.1). Positivity to house dust mite allergen was only associated with patients aged 15 to 20 years (OR = 2.8, 95% CI = 1.0–7.6). The positivity to food allergens by serum test was inversely proportional to the age of the patients and the association was lost between 5 and 10 years (OR = 2.0, 95% CI = 0.7–5.3). There was no statistically significant difference by WHO region and viruses screened subgroups. In metaregression analyses, only the type of LRTI was admitted in the best multivariate model for the type of LRTI in the atopy diagnosed by SPT, atopy diagnosed by serum test, and allergic rhinoconjunctivitis (S6 Table). Follow-up delay of participants was positively associated with house dust mite and negatively associated with atopy diagnosed by serum test for food.

## Confounding factors

A total of 84.8% (89/105) of the 23 confounding factors collected in the included studies had a symmetric distribution between reference cases and control participants (S7 Table). We conducted a sensitivity analysis that included only studies with symmetric distribution for these confounding factors for atopy categories with a significantly different distribution between reference cases and controls (atopy diagnosed by serum tests, positivity to food and inhalant allergens by serum tests, allergic rhinoconjunctivitis, and furred animals). The significant difference observed in the overall analysis was lost in the majority of these categories of atopy (S8 Table).

**Table 2. Atopy development in children with and without lower respiratory tract infections in infancy.**

| | OR (95% CI) | 95% Prediction interval | H¶ (95% CI) | N Studies | N LRTI cases | N controls | I²§ (95%CI) | P heterogeneity | P Egger test |
|---|---|---|---|---|---|---|---|---|---|
| **Atopy diagnosed by SPT** | | | | | | | | | |
| - Overall | 1.2 [0.7–2.0] | [0.1–9.2] | 2.3 [1.9–2.9] | 16 | 1083 | 7651 | 81.8 [71.4–88.3] | < 0.001 | 0,125 |
| - First episode of LRTI | 1.0 [0.6–1.8] | [0.2–5.6] | 1.6 [1–2.6] | 5 | 283 | 6612 | 59.7 [0–85] | 0,042 | 0,193 |
| - Hospitalized | 1.4 [0.7–2.8] | [0.1–16] | 2.4 [1.8–3.1] | 12 | 577 | 827 | 82.3 [70.3–89.4] | < 0.001 | 0,39 |
| - Low risk of bias | 1.2 [0.7–2.1] | [0.1–10] | 2.4 [1.9–3] | 15 | 1046 | 7614 | 82.7 [72.7–89.1] | < 0.001 | 0,076 |
| **Positive serum test** | | | | | | | | | |
| - Overall | 2.0 [1.0–4.1] | [0.2–20.4] | 2.3 [1.6–3.3] | 7 | 636 | 780 | 81.4 [62.5–90.7] | < 0.001 | 0,165 |
| - First episode of LRTI | 2.0 [0.6–6.7] | NA | NA | 1 | 70 | 43 | NA | 1 | NA |
| - Hospitalized | 2.5 [0.9–6.5] | [0.1–87.1] | 2.5 [1.7–3.8] | 5 | 247 | 418 | 84.1 [64.3–92.9] | < 0.001 | 0,188 |
| - Low risk of bias | 2.0 [1.0–4.1] | [0.2–20.4] | 2.3 [1.6–3.3] | 7 | 636 | 780 | 81.4 [62.5–90.7] | < 0.001 | 0,165 |
| **Atopy diagnosis unknown/not reported** | | | | | | | | | |
| - Overall | 0.7 [0.4–1.3] | [0–41.3] | 1 [1–2.6] | 3 | 70 | 99 | 0 [0–84.9] | 0,503 | 0,872 |
| - First episode of LRTI | 1.0 [0.2–4.6] | NA | NA | 1 | 15 | 15 | NA | 1 | NA |
| - Hospitalized | 0.6 [0.3–1.3] | NA | 1 NA | 2 | 55 | 84 | 6.7 NA | 0,301 | NA |
| - Low risk of bias | 0.5 [0.2–1.5] | NA | 1 NA | 2 | 35 | 35 | 5.3 NA | 0,304 | NA |
| **Allergic rhinoconjunctivitis** | | | | | | | | | |
| - Overall | 1.7 [1.1–2.9] | [0.4–7] | 1.6 [1.1–2.3] | 8 | 377 | 607 | 59.3 [11.1–81.3] | 0,016 | 0,615 |
| - First episode of LRTI | 1.4 [0.5–3.6] | NA | 1 NA | 2 | 55 | 60 | 0 NA | 0,993 | NA |
| - Hospitalized | 1.7 [1.1–2.9] | [0.4–7] | 1.6 [1.1–2.3] | 8 | 377 | 607 | 59.3 [11.1–81.3] | 0,016 | 0,615 |
| - Low risk of bias | 1.7 [1.1–2.9] | [0.4–7] | 1.6 [1.1–2.3] | 8 | 377 | 607 | 59.3 [11.1–81.3] | 0,016 | 0,615 |
| **Atopic dermatitis** | | | | | | | | | |
| - Overall | 1.2 [0.9–1.6] | [0.8–1.8] | 1.3 [1–1.7] | 15 | 930 | 1191 | 37.9 [0–66.4] | 0,068 | 0,154 |
| - First episode of LRTI | 1.4 [0.8–2.5] | [0–63.8] | 1 [1–2.6] | 3 | 128 | 133 | 0 [0–85.3] | 0,492 | 0,455 |
| - Hospitalized | 1.2 [0.9–1.6] | [0.8–1.8] | 1.3 [1–1.7] | 15 | 930 | 1191 | 37.9 [0–66.4] | 0,068 | 0,154 |
| - Low risk of bias | 1.3 [0.9–1.8] | [0.6–2.8] | 1.3 [1–1.9] | 13 | 858 | 1090 | 45.1 [0–71.3] | 0,039 | 0,224 |
| **Pollens** | | | | | | | | | |
| - Overall | 0.8 [0.3–2.7] | [0–773146.5] | 2.3 [1.3–4.1] | 3 | 134 | 223 | 81.6 [42.8–94.1] | 0,004 | 0,214 |
| - Hospitalized | 0.8 [0.3–2.7] | [0–773146.5] | 2.3 [1.3–4.1] | 3 | 134 | 223 | 81.6 [42.8–94.1] | 0,004 | 0,214 |

(*Continued*)

**Table 2.** (Continued)

| | OR (95% CI) | 95% Prediction interval | H¶ (95% CI) | N Studies | N LRTI cases | N controls | I²§ (95%CI) | P heterogeneity | P Egger test |
|---|---|---|---|---|---|---|---|---|---|
| - Low risk of bias | 0.8 [0.3–2.7] | [0–773146.5] | 2.3 [1.3–4.1] | 3 | 134 | 223 | 81.6 [42.8–94.1] | 0,004 | 0,214 |
| **Food allergy** | | | | | | | | | |
| - Overall | 0.8 [0.3–2.5] | [0–26.7] | 1.3 [1–2.3] | 4 | 195 | 292 | 43.2 [0–81] | 0,152 | 0,297 |
| - First episode of LRTI | 0.8 [0.2–3.8] | NA | 1 NA | 2 | 55 | 60 | 0 NA | 0,814 | NA |
| - Hospitalized | 0.8 [0.3–2.5] | [0–26.7] | 1.3 [1–2.3] | 4 | 195 | 292 | 43.2 [0–81] | 0,152 | 0,297 |
| - Low risk of bias | 0.8 [0.3–2.5] | [0–26.7] | 1.3 [1–2.3] | 4 | 195 | 292 | 43.2 [0–81] | 0,152 | 0,297 |
| **Furred animals** | | | | | | | | | |
| - Overall | 0.6 [0.5–0.9] | [0.4–0.9] | 1 [1–1.7] | 9 | 382 | 7179 | 1 [0–65.1] | 0,426 | 0,152 |
| - First episode of LRTI | 0.5 [0.3–1.0] | [0–44.7] | 1 [1 – 1] | 3 | 103 | 6674 | 0 [0–0] | 0,913 | 0,004 |
| - Hospitalized | 0.6 [0.5–0.9] | [0.4–0.9] | 1.1 [1–1.5] | 8 | 334 | 565 | 13.3 [0 – 56] | 0,326 | 0,1 |
| - Low risk of bias | 0.6 [0.5–0.9] | [0.4–0.9] | 1 [1–1.7] | 9 | 382 | 7179 | 1 [0–65.1] | 0,426 | 0,152 |
| **House dust mite** | | | | | | | | | |
| - Overall | 1.1 [0.6–2.2] | [0.2–6.8] | 1.4 [1–2.4] | 5 | 255 | 6961 | 52 [0–82.4] | 0,08 | 0,552 |
| - First episode of LRTI | 0.7 [0.3–1.6] | NA | 1 NA | 2 | 122 | 6738 | 0 NA | 0,385 | NA |
| - Hospitalized | 1.6 [0.8–2.9] | [0.4–6] | 1.3 [1–2.2] | 4 | 206 | 296 | 41.1 [0–80.1] | 0,165 | 0,032 |
| - Low risk of bias | 1.1 [0.6–2.2] | [0.2–6.8] | 1.4 [1–2.4] | 5 | 255 | 6961 | 52 [0–82.4] | 0,08 | 0,552 |
| **Positive serum test Food** | | | | | | | | | |
| - Overall | 5.3 [1.7–16.7] | [0–1015892.4] | 1.9 [1–3.5] | 3 | 130 | 263 | 71.5 [3.4–91.6] | 0,03 | 0,369 |
| - Hospitalized | 5.3 [1.7–16.7] | [0–1015892.4] | 1.9 [1–3.5] | 3 | 130 | 263 | 71.5 [3.4–91.6] | 0,03 | 0,369 |
| - Low risk of bias | 5.3 [1.7–16.7] | [0–1015892.4] | 1.9 [1–3.5] | 3 | 130 | 263 | 71.5 [3.4–91.6] | 0,03 | 0,369 |
| **Positive serum test Inhalants** | | | | | | | | | |
| - Overall | 4.2 [2.1–8.5] | [0–402.8] | 1.1 [1–3.3] | 3 | 130 | 263 | 10.5 [0–90.7] | 0,327 | 0,814 |
| - Hospitalized | 4.2 [2.1–8.5] | [0–402.8] | 1.1 [1–3.3] | 3 | 130 | 263 | 10.5 [0–90.7] | 0,327 | 0,814 |
| - Low risk of bias | 4.2 [2.1–8.5] | [0–402.8] | 1.1 [1–3.3] | 3 | 130 | 263 | 10.5 [0–90.7] | 0,327 | 0,814 |

SPT: Skin prick test; LRTI: Lower respiratory tract infections; N: Number; 95% CI: 95% Confidence Interval; NA: Not Applicable

¶H is a measure of the extent of heterogeneity, a value of H = 1 indicates homogeneity of effects and a value of H >1indicates a potential heterogeneity of effects.

§: I2 describes the proportion of total variation in study estimates that is due to heterogeneity, a value > 50% indicates presence of heterogeneity

## Heterogeneity and publication bias analysis

There was no heterogeneity in overall and sensitivity analyses for atopy with unknown or not reported diagnosis method, atopic dermatitis, food allergy, furred animals, and positive serum

tests for inhalants (Table 2). The analyses of atopy diagnosed by SPT showed a publication bias (P Egger = 0.085). The funnel diagrams of the main analysis are presented in the S2–S12 Figs.

## Discussion

Our results highlight that there is no relationship between a history of LRTI at < 5 years and atopy diagnosed by SPT, atopy diagnosed unknown/not reported, atopic dermatitis, and hyperresponsiveness to common allergens including pollen, food allergens or house dust mites. Our results on atopy diagnosed by serum tests, allergic rhinoconjunctivitis, and positivity by serum tests to food or inhaled allergens cannot be definite with an increased risk observed in the global analysis and not confirmed by the analyses in studies reporting confounding factors. The increased risks of developing atopy observed in some subgroup analyses were more frequent in case of bronchiolitis due to HRSV between 9–12 months and in prospective studies conducted in Europe.

Our results are consistent with the quantitative analysis by Knyber et al. who concluded that there was no relationship between hospitalization for HRSV bronchiolitis at < 1 year and subsequent allergic sensitization [10]. Similar to the findings of this review, Kneyber et al. also concluded that HRSV infection in childhood was associated with allergic sensitization to food or inhaled allergens tested with serum tests. However, we have no definitive conclusion on this point since our analyses, taking into account studies with confounders, such as family history of atopy [51], did not confirm this finding. Similar to the findings of the present work, several studies have also shown divergent results between serum and skin tests [52–55]. There are many hypothetical reasons that may explain these observed differences between serum and skin test results. First, differences in the composition and/or concentration of skin and serum tests targets may lead to differences in the results of both tests [52]. In a context of immune immaturity, for example, insufficient migration of mast cells to the epidermis could lead to positive results for serum tests and false negative for skin tests. Serum tests also involve false positive results due to nonspecific binding with the antibodies used [56]. Technical differences in the handling of skin and serum tests may also be involved in the differences observed between the two methods [57]. The systematic review by Fauroux et al. reported for studies conducted between 1995 and 2015 in industrialized countries the controversial nature of the results on the association between infantile hospitalizations for HRSV LRTI and subsequent atopy [58]. Pérez-Yarza in a systematic review including children younger than 3 years with HRSV respiratory infection from 1985 to 2006 also suggested controversial findings about the subsequent risk of allergic sensitization development defined by positive skin or serum tests specific for common allergens [11].

### Strengths and limitations

While this systematic review may help clarify the relationship between LRTI in childhood and subsequent atopy, the weaknesses of the work must be emphasized. More than three quarters of the included studies in this systematic review were from Europe. This suggests an important problem in the external validity of our results on a global scale, with the absence of America, South East Asia and Eastern Mediterranean.

This systematic review is the only one to date to address this topic with a strict atopy definition with the consideration of 11 different categories depending on the type of diagnosis used, allergic diseases and sensitization to common allergens. This systematic review includes a multitude of sensitivity analyses with studies reporting their first episode of LRTI, studies reporting children hospitalized for LRTI, and quality of studies. Other special strengths of this systematic review include the large size of the participants included, 5294 reference cases and

27091 controls, the long follow-up period of more than half a century of children from birth to about 30 years old and with several points of follow-up including all age groups. The data was carefully extracted from a structured questionnaire and we used an appropriate data analysis to consider 23 important confounding variables.

## Conclusions

No relationship was found in this systematic review between viral LRTI at <5 years and the subsequent development of a SPT-diagnosed atopy, sensitization to common allergens or the development of atopic dermatitis. This conclusion was not confirmed for the association between viral LRTI at < 5 years and the subsequent development of serum test diagnosed atopy, serum test positive for food or inhaled allergens, allergic rhinoconjunctivitis, and sensitization to furred animals.

Thus, more longitudinal investigations adjusted to confounding factors are important to elucidate the implication of childhood LRTI in the development of atopy or allergy to food or inhalant assessed by serum tests, allergic rhinoconjunctivitis, and sensitization to furred animals. These findings should encourage research on the long-term burden of viral LRTI in childhood in non-European regions and non-HRSV viruses. Prospective randomized studies including intervention against the development of the LRTI would be ideal to rule out the residual confusion about the causal relationship between infantile LRTIs and the development of subsequent atopy. The imminent arrival of the vaccine against HRSV on the market or the prophylactic means such as palivizumab could be a way to carry out these interventional studies. To reduce the burden of atopy, there a real need of more accurate diagnosis tools and efforts should focus on other major risk factors including genetic predisposition, diet habits, air pollution, family size, and the use of vaccines or antibiotics.

## Supporting information

**S1 Table. PRISMA 2009 checklist.**
(PDF)

**S2 Table. Items for risk of bias assessment.**
(PDF)

**S3 Table. Individual characteristics of included studies.**
(PDF)

**S4 Table. Risk of bias assessment.**
(PDF)

**S5 Table. Subgroup analyses of atopy in children with LRTI in infancy and control without respiratory diseases.**
(PDF)

**S6 Table. Metaregression analyses for the association of LRTI with subsequent atopy.**
(PDF)

**S7 Table. P-value of Khi-2 and Fisher exact tests for qualitative confounding factors.**
(PDF)

**S8 Table. Sensitivity analyses of the symmetrically distributed confounding factors.**
(PDF)

**S1 Fig. Forest plot of the comparison of atopy in people with and without LRTI in infancy.**
(PDF)

**S2 Fig. Funnel plot for publication for atopy diagnosed by skin prick tests.**
(PDF)

**S3 Fig. Funnel plot for publication for atopy diagnosed by serum tests.**
(PDF)

**S4 Fig. Funnel plot for publication for atopy diagnosis unknown/not reported.**
(PDF)

**S5 Fig. Funnel plot for publication for allergic rhinoconjunctivitis.**
(PDF)

**S6 Fig. Funnel plot for publication for atopic dermatitis.**
(PDF)

**S7 Fig. Funnel plot for publication for pollens.**
(PDF)

**S8 Fig. Funnel plot for publication for food allergy.**
(PDF)

**S9 Fig. Funnel plot for publication for furred animals.**
(PDF)

**S10 Fig. Funnel plot for publication for house dust mite.**
(PDF)

**S11 Fig. Funnel plot for publication for positive serum test for food.**
(PDF)

**S12 Fig. Funnel plot for publication for positive serum test for inhalants.**
(PDF)

## Author Contributions

**Conceptualization:** Sebastien Kenmoe, Richard Njouom.

**Data curation:** Sebastien Kenmoe, Cyprien Kengne-Nde.

**Formal analysis:** Sebastien Kenmoe, Cyprien Kengne-Nde.

**Methodology:** Sebastien Kenmoe, Cyprien Kengne-Nde, Abdou Fatawou Modiyinji, Jean Joel Bigna, Richard Njouom.

**Project administration:** Sebastien Kenmoe, Richard Njouom.

**Software:** Sebastien Kenmoe, Cyprien Kengne-Nde.

**Supervision:** Sebastien Kenmoe, Richard Njouom.

**Validation:** Sebastien Kenmoe, Abdou Fatawou Modiyinji, Jean Joel Bigna, Richard Njouom.

**Writing – original draft:** Sebastien Kenmoe.

**Writing – review & editing:** Sebastien Kenmoe, Cyprien Kengne-Nde, Abdou Fatawou Modiyinji, Jean Joel Bigna, Richard Njouom.

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
