## [Decision Letter · Decision Letter 0]

9 Dec 2019

PONE-D-19-27663

Association of early viral lower respiratory infections and subsequent development of atopy, a systematic review and meta-analysis of cohort studies

PLOS ONE

Dear PhD Njouom,

Thank you for submitting your manuscript to PLOS ONE. After careful consideration, we feel that it has merit but does not fully meet PLOS ONE’s publication criteria as it currently stands. Therefore, we invite you to submit a revised version of the manuscript that addresses the points raised during the review process.

We would appreciate receiving your revised manuscript by Jan 23 2020 11:59PM. To enhance the reproducibility of your results, we recommend that if applicable you deposit your laboratory protocols in protocols.io, where a protocol can be assigned its own identifier (DOI) such that it can be cited independently in the future. For instructions see: http://journals.plos.org/plosone/s/submission-guidelines#loc-laboratory-protocols

We look forward to receiving your revised manuscript.

Kind regards,

Renee W.Y. Chan, Ph.D.

Academic Editor

PLOS ONE

Journal Requirements:

1)

2) Thank you for stating the following financial disclosure:

Please provide an amended Funding Statement that declares *all* the funding or sources of support received during this specific study (whether external or internal to your organization) as detailed online in our guide for authors at http://journals.plos.org/plosone/s/submit-now.  Please state what role the funders took in the study.  If any authors received a salary from any of your funders, please state which authors and which funder. If the funders had no role, please state: "The funders had no role in study design, data collection and analysis, decision to publish, or preparation of the manuscript."

3) Please include captions for your Supporting Information files at the end of your manuscript, and update any in-text citations to match accordingly. Please see our Supporting Information guidelines for more information: http://journals.plos.org/plosone/s/supporting-information.

Reviewers' comments:

Reviewer's Responses to Questions

**Comments to the Author**

1. Is the manuscript technically sound, and do the data support the conclusions?

Reviewer #1: No

Reviewer #2: Yes

Reviewer #3: Yes

2. Has the statistical analysis been performed appropriately and rigorously? 

Reviewer #1: No

Reviewer #2: Yes

Reviewer #3: Yes

3. Have the authors made all data underlying the findings in their manuscript fully available?

Reviewer #1: Yes

Reviewer #2: Yes

Reviewer #3: Yes

4. Is the manuscript presented in an intelligible fashion and written in standard English?

Reviewer #1: No

Reviewer #2: Yes

Reviewer #3: Yes

5. Review Comments to the Author

Reviewer #1: The authors choose an interesting but very challenging topic, unfortunately their strategy has some serious flaws which make this paper requires intensive amendment.

1. Abstract: Conclusion “These results suggest that there is no association between viral LRTI at <5 years and the subsequent atopy”. but the results suggest there is an association?

2. It is not clear why the authors chose this research question. The authors may consider adding more background information about the potential linkage of viral LRTI in early childhood and atopy. To me, this association could be due to immature immune of children, so they are more likely to have both viral infections and allergies, which may not indicate the infections increase the chance of allergy. Also, why did they only include viral LRTI? We know bacterial LRTI are very common in children too. The age cut-off for LRTI is 5yrs, but why is atopy set to >2yrs? The more intuitive definition could be incidence of atopy after the first episode of LRTI.

3. Their search strategy should be moved into main text since it is critical for a systematic review. The authors declare “Limit #3 to humans”, how did they achieve it? by screening title and abstract? or include keywords related to human? The keyword combinations should be standardised, eg. “Atopy OR allergy OR hypersensitivity OR allerg* OR atop* OR asthma OR asthma* OR “immunoglobulin E” OR “Ig E”, why quotation marks are added to some of keywords? also atop* includes atopy, and IgE is a standard term instead of Ig E. The title is viral LRTI but there is no keywords specifically for viral infections. Influenza is one of most common causes of viral infections but apparently excluded by the authors for no reasons. Hence I really doubt this strategy could have missed a lot of related papers. It is surprising to see no keywords related to children <2 yrs or >5yrs and cohort study design.

4. They authors did not define viral LRTI clearly, and total rely on the terms adopted by individual studies. Was it based on symptoms, clinical records, or has to be confirmed by lab results? If the last one, it should be noted that few studies have tested every participant for viral infections, so these cases could be seriously under-reported.

5. Table 1. It is more conventional to summarise study by study. Also I am surprising to see none of the selected studies were from the US and none tested for flu (because influenza was not included in search keywords?)

6. L72. No need to add “human” to every virus, since the studies were conducted in children.

7. L96. “Children with 97 viral LRTI, reference cases, were compared to control children who had no history of LRTI in childhood”. add “as” before reference cases to avoid confusion.

8. Supplemental Table 3. This table does not provide useful information. NOS score of each selected study should be added to the summary table.

9. Supplemental table 4 is not necessary.

10. L145, 146. what are Khi-2 and Chi-2? Both are Chi-square tests?

Reviewer #2: This paper investigates the association between viral LRT1 in childhood and subsequent development of atopy through systematic review.

Comments:

There should be a discussion about the possible reason for the different results obtained by SPT and serum test.

Page 4, line 77-78, why opposite meaning to the previous one “conversely”?

Page 7, line 145 “Khi-2” – Chi-sq?

Reviewer #3: The present is an interesting meta-analysis aiming to evaluate relationship between LRTI and atopy in 5 years

The paper is well written and was recorded on PROSPERO.

Abstract,

It should be added if relationship was evalauted with multivariate model or not

Methods.

Random effect was correctly choosen. It should be added if data derived came from multivariate analysis or not.

Meta regression for age and lenght of follow up should be added

6. PLOS authors have the option to publish the peer review history of their article (what does this mean?). If published, this will include your full peer review and any attached files.

Reviewer #1: No

Reviewer #2: No

Reviewer #3: Yes: Fabrizio D'Ascenzo

---

## [Author Response · Author response to Decision Letter 0]

18 Jan 2020

Reviewer #1: The authors choose an interesting but very challenging topic, unfortunately their strategy has some serious flaws which make this paper requires intensive amendment.

1. Abstract: Conclusion “These results suggest that there is no association between viral LRTI at <5 years and the subsequent atopy”. but the results suggest there is an association?

Authors: We thank the reviewer for this relevant comment. The conclusion has been modified and can now be read in the revised version as: "These results suggest that there is no association between LRTI at <5 years and the majority of categories of atopy studied in this work. This result, however, is not confirmed for the remaining categories of atopy and more particularly those diagnosed by serum tests. There is a real need to develop more accurate atopy diagnostic tools."

2. It is not clear why the authors chose this research question. The authors may consider adding more background information about the potential linkage of viral LRTI in early childhood and atopy. To me, this association could be due to immature immune of children, so they are more likely to have both viral infections and allergies, which may not indicate the infections increase the chance of allergy.

Authors: We thank the reviewer for these thoughtful comments. A paragraph that clarifies the association between viral infections in childhood, primarily HRSV and RV, and the development of wheezing/asthma later was added in background. Data on the association between these viral LRTIs in childhood and the development of atopy later on contrary have remained conflictual to date. This is why we started the present work to determine the link between viral LRTI in childhood and the development of atopy later.

Also, why did they only include viral LRTI? We know bacterial LRTI are very common in children too. 

Authors: Dear reviewer, thank you for this important comment. A first systematic review and meta-analysis has already been conducted on the long-term sequelae of pneumonia, including bacteria pneumonia (3 studies with Mycoplasma pneumoniae, 1 with Staphlococcus aureus, and 1 with Chlamydia pneumoniae) (Edmond et al., 2012, 10.1371 / journal.pone.0031239). The subgroup analyses in this meta-analysis had shown that the 3 included studies with Mycoplasma pneumoniae pneumonia were not associated with long-term sequelae. 

The authors agree with the reviewer that bacteria are also responsible for childhood LRTI and could therefore be associated with the development of atopy later. However, this was not the aim of the present study. We aimed at investigating the association between childhood LRTIs with laboratory confirmed viral infection and the subsequent atopy development. This objective still does not exclude the possibility of viral and bacterial co-infections in the studies that have been included. However, the majority of the authors of the included studies did not report the coinfections, so it is difficult for us to be able to discuss the contribution of bacteria in the effect reported in the present study.

The age cut-off for LRTI is 5yrs, but why is atopy set to >2yrs? The more intuitive definition could be incidence of atopy after the first episode of LRTI.

Authors: We thank the reviewer for this comment. We chose LRTI at <5 years since this is the most at risk age group for LRTI infections. Also, atopy at> 2 years is not link to any issue, since we only consider studies with the episode of atopy occurring after that of LRTI. We also have only 6 out of the 24 included studies that confirmed that participants were presenting the first episode of LRTI. We conducted sensitivity analyses with studies with participants having the first episode of LRTI and no difference in effect was observed compared to the overall results. These are all reasons why we keep our inclusion criteria without change.

3. Their search strategy should be moved into main text since it is critical for a systematic review. 

Authors: The search strategy is now presented in Table 1 of the main manuscript as suggested by the reviewer. Thanks for the comment. 

The authors declare “Limit #3 to humans”, how did they achieve it? by screening title and abstract? or include keywords related to human? 

Authors: Thanks to the reviewer for these comments. Unfortunately, we inadvertently submitted the article with an old version of the search strategy that we developed. We have included the final search strategy that does not contain this filter in the current version of the manuscript. This is actually a filter that is available by default in the Pubmed database that we used during the testing phase of our search strategies.

The keyword combinations should be standardised, eg. “Atopy OR allergy OR hypersensitivity OR allerg* OR atop* OR asthma OR asthma* OR “immunoglobulin E” OR “Ig E”, why quotation marks are added to some of keywords? also atop* includes atopy, and IgE is a standard term instead of Ig E. 

Author: The final search strategy has been standardized as recommended by the reviewer, thanks for the comment.

The title is viral LRTI but there is no keywords specifically for viral infections. Influenza is one of most common causes of viral infections but apparently excluded by the authors for no reasons. Hence I really doubt this strategy could have missed a lot of related papers. It is surprising to see no keywords related to children <2 yrs or >5yrs and cohort study design.

Authors: The final research strategy that included the specific keywords of the study's major areas of interest is now clarified in the manuscript (LRTI, virus and atopy). We opted for a very sensitive search strategy to increase our chances of not missing relevant articles. The included studies were longitudinal and some followed patients even until the age of 30. We therefore considered appropriate to not restrict the age of the participants according to our inclusion criteria (<2 years or <5 years). Similarly, the study design specific keywords are not always reported by some authors reason why we did not associate restriction according to this criterion in our search strategy.

4. They authors did not define viral LRTI clearly, and total rely on the terms adopted by individual studies. Was it based on symptoms, clinical records, or has to be confirmed by lab results? If the last one, it should be noted that few studies have tested every participant for viral infections, so these cases could be seriously under-reported.

Authors: We thank reviewer for the comment. The 24 included studies in the present systematic review confirmed laboratory viral infection in all participants. This was the main eligibility criterion. We have now added individual LRTI case definitions for each study included in the supplementary table 4 of individual data of included studies. Case definitions were based on clinical symptoms recorded in hospitals prospectively or in secure databases or radiographic exams.

5. Table 1. It is more conventional to summarise study by study. 

Authors: We thank reviewer for the comment. We have now removed Table 1. The supplementary table 4 of individual data of included studies is now to be considered for the description of included studies.

Also I am surprising to see none of the selected studies were from the US and none tested for flu (because influenza was not included in search keywords?)

Authors: Only two included studies had performed the detection of common respiratory viruses including Influenza virus (Nicolai, 2017 et al., doi: 10.1097/INF.0000000000001385 and Ruotsalainen et al., 2013, doi: 10.1002/ppul.22692). Indeed, to date most studies on the long-term sequelae of LRTI in childhood have focused mainly on the involvement of HRSV bronchiolitis in the development of subsequent wheezing or asthma. Authors agree with the reviewer that many of the eligible studies examined were conducted in the US but none of these studies met the inclusion criteria of the present article.

6. L72. No need to add “human” to every virus, since the studies were conducted in children.

Authors: We thank reviewer for this comment, the text is now modified accordingly.

7. L96. “Children with 97 viral LRTI, reference cases, were compared to control children who had no history of LRTI in childhood”. add “as” before reference cases to avoid confusion.

Authors: We thank reviewer for this comment, the text is now modified accordingly.

8. Supplemental Table 3. This table does not provide useful information. NOS score of each selected study should be added to the summary table.

Authors: A supplementary table 3 that specifying the NOS scale ratings has been added to the appendix.

9. Supplemental table 4 is not necessary.

Authors: Thank you for the comment, the supplementary table 4 has been removed.

10. L145, 146. what are Khi-2 and Chi-2? Both are Chi-square tests?

Authors: Yes both are Chi-square tests, the manuscript is now corrected accordingly.

Reviewer #2: This paper investigates the association between viral LRT1 in childhood and subsequent development of atopy through systematic review.

Comments:

There should be a discussion about the possible reason for the different results obtained by SPT and serum test.

Authors: We really appreciate the suggestion. The following paragraph has been added in discussion section. "Similar to the findings of the present work, several studies have also shown divergent results between serum and skin tests [1–4]. There are many hypothetical reasons that may explain these observed differences between serum and skin test results. First, differences in the composition and/or concentration of skin and serum tests targets may lead to differences in the results of both tests [1]. In a context of immune immaturity, for example, insufficient migration of mast cells to the epidermis could lead to positive results for serum tests and false negative for skin tests. Serum tests also involve false positive results due to nonspecific binding with the antibodies used [5]. Technical differences in the handling of skin and serum tests may also be involved in the differences observed between the two methods [6]."

Page 4, line 77-78, why opposite meaning to the previous one “conversely”?

Authors: "conversely" has been removed, thank for the comment.

Page 7, line 145 “Khi-2” – Chi-sq?

Authors: Both are Chi-square tests; the manuscript is now corrected accordingly.

Reviewer #3: The present is an interesting meta-analysis aiming to evaluate relationship between LRTI and atopy in 5 years

The paper is well written and was recorded on PROSPERO.

Abstract,

It should be added if relationship was evalauted with multivariate model or not

Authors: Thank you to the reviewer for this suggestion. We have now specified in the abstract that we have performed multivariate metaregresssion. 

Methods.

Random effect was correctly choosen. It should be added if data derived came from multivariate analysis or not.

Authors: We performed multivariate analyzes only in metaregression. However, we have not obtained any multivariate metaregression model including two or more factors associated with the development of atopy following LRTI.

Meta regression for age and lenght of follow up should be added

Authors: Thanks to the reviewers for this suggestion, we have now done a metaregression to find the factors linked to the development of atopy following childhood LRTI. We have added the corresponding methodology and the results obtained (Supplementary Table 6). We only considered the length of follow-up of the children in the model since it represents the difference between the age at the end and at the start of follow-up.

---

## [Decision Letter · Decision Letter 1]

9 Mar 2020

PONE-D-19-27663R1

Association of early viral lower respiratory infections and subsequent development of atopy, a systematic review and meta-analysis of cohort studies

PLOS ONE

Dear PhD Njouom,

Thank you for submitting your manuscript to PLOS ONE. After careful consideration, we feel that it has merit but does not fully meet PLOS ONE’s publication criteria as it currently stands. Therefore, we invite you to submit a revised version of the manuscript that addresses the points raised during the review process.

Please address the concern raised by Reviewer 3.

We would appreciate receiving your revised manuscript by Apr 23 2020 11:59PM. To enhance the reproducibility of your results, we recommend that if applicable you deposit your laboratory protocols in protocols.io, where a protocol can be assigned its own identifier (DOI) such that it can be cited independently in the future. For instructions see: http://journals.plos.org/plosone/s/submission-guidelines#loc-laboratory-protocols

We look forward to receiving your revised manuscript.

Kind regards,

Renee W.Y. Chan, Ph.D.

Academic Editor

PLOS ONE

Reviewers' comments:

Reviewer's Responses to Questions

**Comments to the Author**

1. If the authors have adequately addressed your comments raised in a previous round of review and you feel that this manuscript is now acceptable for publication, you may indicate that here to bypass the “Comments to the Author” section, enter your conflict of interest statement in the “Confidential to Editor” section, and submit your "Accept" recommendation.

Reviewer #1: All comments have been addressed

Reviewer #2: All comments have been addressed

Reviewer #3: (No Response)

2. Is the manuscript technically sound, and do the data support the conclusions?

Reviewer #1: Yes

Reviewer #2: Yes

Reviewer #3: Yes

3. Has the statistical analysis been performed appropriately and rigorously? 

Reviewer #1: Yes

Reviewer #2: Yes

Reviewer #3: No

4. Have the authors made all data underlying the findings in their manuscript fully available?

Reviewer #1: Yes

Reviewer #2: Yes

Reviewer #3: Yes

5. Is the manuscript presented in an intelligible fashion and written in standard English?

Reviewer #1: Yes

Reviewer #2: Yes

Reviewer #3: Yes

6. Review Comments to the Author

Reviewer #1: (No Response)

Reviewer #2: The manuscript is much improved after the revision. The authors have addressed all my questions. I have no further comments.

Reviewer #3: The most relevant problem there is the level of evidence. That is the measure of association do not derive from multivariate analysis of original studies but from multivariate metaregression which is totally different. This should be' clearly stated.

7. PLOS authors have the option to publish the peer review history of their article (what does this mean?). If published, this will include your full peer review and any attached files.

Reviewer #1: No

Reviewer #2: No

Reviewer #3: Yes: Fabrizio D'Ascenzo

---

## [Author Response · Author response to Decision Letter 1]

9 Mar 2020

Reviewer #1: (No Response)

We thank the reviewers for their relevant comments.

Reviewer #2: The manuscript is much improved after the revision. The authors have addressed all my questions. I have no further comments.

We thank the reviewers for their relevant comments.

Reviewer #3: The most relevant problem there is the level of evidence. That is the measure of association do not derive from multivariate analysis of original studies but from multivariate metaregression which is totally different. This should be' clearly stated.

We thank to the reviewer for this suggestion. We had previously specified in the abstract that we had conducted multivariate metaregression (Page 2, Line 30). We have now precise also in Methods section that we have performed multivariate metaregression (Page 8, Line 157).

---

## [Editor Report · Decision Letter 2]

2 Apr 2020

Association of early viral lower respiratory infections and subsequent development of atopy, a systematic review and meta-analysis of cohort studies

PONE-D-19-27663R2

Dear Dr. Njouom,

We are pleased to inform you that your manuscript has been judged scientifically suitable for publication and will be formally accepted for publication once it complies with all outstanding technical requirements.

With kind regards,

Renee W.Y. Chan, Ph.D.

Academic Editor

PLOS ONE
---

## [Editor Report · Acceptance letter]

8 Apr 2020

PONE-D-19-27663R2 

Association of early viral lower respiratory infections and subsequent development of atopy, a systematic review and meta-analysis of cohort studies 

Dear Dr. Njouom:

I am pleased to inform you that your manuscript has been deemed suitable for publication in PLOS ONE. Congratulations! Your manuscript is now with our production department. 

With kind regards,

on behalf of

Dr. Renee W.Y. Chan 

Academic Editor

PLOS ONE